mathematical modelling/complexity/applied mathematics

social costs, sustainable mobility, traffic, cars

**Author for correspondence:**
Rafael Prieto Curiel
e-mail: rafael.prieto.13@ucl.ac.uk

# A paradox of traffic and extra cars in a city as a collective behaviour

Rafael Prieto Curiel[1], Humberto González Ramírez[2,3], Mauricio Quiñones Domínguez[4] and Juan Pablo Orjuela Mendoza[5]

[1]Centre for Advanced Spatial Analysis, University College London, Gower Street, London WC1E 6BT, UK
[2]Transport and Traffic Engineering Laboratory (LICIT), University Gustave Eiffel, Marne la Vallee, France
[3]Transport and Traffic Engineering Laboratory (LICIT), University Lyon, ENTPE, Lyon, France
[4]Research in Spatial Economics, Universidad EAFIT, Medellín, Antioquia, Colombia
[5]Transport Studies Unit, School of Geography and the Environment, University of Oxford, South Parks Road, Oxford, UK

RPC, 0000-0002-0738-2633; HGR, 0000-0002-3988-534X; MQD, 0000-0002-3696-6865

Promoting walking or cycling and reducing cars' use is one of the city planners' main targets, contributing to a sustainable transport method. Yet, the number of vehicles worldwide is increasing as fast as the population, and motorized mobility has become the primary transport method in most cities. Here, we consider modal share as an emergent behaviour of personal decisions. All individuals minimize their commuting time and reach an equilibrium under which no person is willing to change their transportation mode. In terms of the minimum travel time, the best-case scenario is used to determine the extra commuting time and the excess cars, computed as a social inefficiency. Results show that commuting times could increase up to 25% with many more vehicles than optimum. Paradoxically, all individuals trying to minimize their time could collectively reach the maximum commuting times in the extreme case, with all individuals driving during rush hour.

## 1. Introduction

Designing an adequate system for urban mobility plays a crucial role in the sustainability of our planet. With more than 80 million cars being produced worldwide each year, manufacturing vehicles roughly contributes 4% of the total carbon dioxide emissions. In cities, cars require expensive infrastructure and take

much space, usually in very inefficient ways. Increasing active transport, such as walking or cycling, is proposed to address both health and environmental issues [1–4] and public transport provides a less polluting, low-cost and socially equitable travel alternative [5]. For every driver that switches to other transportation modes, such as walking, cycling or public transport, many daily kilos of $CO_2$ emissions are saved [6]. Yet, convincing drivers to switch to other transportation modes is a big challenge in most cities, where commuters are inclined to use a private car, imposing negative costs to the rest of the city, such as crashes and road injuries, congestion, air pollution, noise and the inefficient use of public space [7,8]. Promoting sustainable transport is the objective of many cities and a core part of one of the Sustainable Development Goals (11.2 and 11.6), where the main goal is that more trips are non-motorized (walking and cycling) or using public transport.

In many cities, a large part of their trips are motorized, and the non-motorized trips or the use of public transport are still not desirable systems for commuting. For example, the percentage of trips made by car in cities of more than 4000 inhabitants per square kilometre is more than 70% in the USA and 45% in Germany [9], with this percentage growing to more than 90% in some US cities like Detroit, Houston and Dallas [10]. The reasons for the car-based mobility are many, including comfort, perception of more security when using a car, fixed schedules, scarce availability of public transport stations or the expected extra travel time and more [11], and for a city planner, it is not easy to tackle them all [12]. Furthermore, with an increased number of vehicles in the city, policymakers are inclined to construct even more car infrastructure and invest even more in private cars, which then creates more incentives for private vehicle use and results in even more congestion [11,13]. A recent study [14] showed that in Mexico City, nearly 18% of the people would use a car (or taxi or motorbike) if they could afford it [15], meaning that money is one of the main restrictions that prevent the number of cars from growing even faster [16]. Thus, it is likely that as the income in Mexico City increases, more people will decide to use a car as an option [17–19]. Similar issues are being faced among other Latin American cities, such as Bogotá and Lima, and will be faced as motorization increases in Bangalore, Manila, Jakarta and others.

Motorization is a complex phenomenon, where citizens decide based on individual preferences, but the system, formed by the rest of the users (in the way of traffic jams or the saturation of the transport system) is the result of a collective, yet not fully understood behaviour, with nonlinear impacts and feedbacks between the system and the population. The number of people who use a private car, and the resulting social inefficiencies, is a collective decision-making process that emerges in a city, meaning that it can only be understood when all individuals decide whether to use a car and interact with others.

Here, we analyse the car-based mobility of a city from the perspective of emergent behaviours of individuals who choose to commute using either a car or anything else, including public transport, cycling and walking. The interaction between drivers and public transport users, and among themselves, is quantified as direct or induced costs, measured as commuting time. As a sequential decision process with perfect information (that is, users are aware of the expected time of using either system), a collective dynamic is considered, in which users try to minimize their costs. If equilibrium is reached, we compare their average commuting time and the extra number of cars in the city against the best-case scenario, in which the share between car users and non-car users minimizes the mean commuting time. Results of our model show that social inefficiency is reached and very common among users. As expected, more public transport infrastructure (or more cycling lanes and wider sidewalks, for example) would reduce its baseline costs and increase its number of users. Yet, with the same infrastructure and baseline costs, it is possible to encourage public transport, if the interaction between the two modes of transportation is more prominent. The average commuting time could be shortened by up to 25% with a non-selfish modal share in the city, and the number of cars could be reduced considerably as well. In the extreme case, we find a paradox, where all individuals decide their commuting mode trying to minimize their costs, but the emergent result is the overall worst-case scenario, where the average commuting time is maximum and where all people decide to use their car.

## 2. Modelling car-based mobility as a collective behaviour

Modelling transport mode choice has been a topic of interest for transport planners and researchers for decades [20–24]. One of the most influential techniques is the four-stage transport model, which uses spatial data to identify trip generation and attraction for each spatial unit, distributes such trips and then allocates a modal split based on some travel costs [25]. However, these models require extensive spatial data that is not always available. Still, more straightforward methodologies could shed light on

policy implications without the need for detailed spatial data. The use of simple mathematical relations can help understand motorization in cities and offer an additional tool for designing transport interventions and mobility policies.

## 2.1. Choice of the mode of transportation

Travellers' choice behaviour is a process that involves psychological and cognitive mechanisms through which they perceive the states of the world and then make decisions accordingly [26,27]. Many factors intervene in this process [9], including characteristics related to the trip, such as time of the day, the purpose of the trip and its length, but also attributes of each mode of transportation and the number of options available for the user, including monetary costs, how consistent and predictable travel times are, parking availability, walking distance, waiting time, frequency, number of changes, the level of satisfaction, comfort and security [28–30]. Beyond trip and travel factors, other aspects such as socio-economic and demographic characteristics of the travellers, including income, car or bike ownership, gender, age, social status and environmental and health consciousness, also play a significant role [19,31–35], as well as past experiences with a mode of transport [28], transport and land-use policies, urban sprawl, subsidies, car use restrictions, public transport infrastructure, and level of service or public transport systems [11,32,33,35–38]. Even weather conditions play a role [39,40] on whether a person chooses to drive instead of using the bus. These factors determine the cost incurred by travellers when selecting a mode of transportation. Car or public transport ridership is driven by the users' perceived cost [41–44].

In transportation, as well as in economics, several theories have been proposed to explain how individuals make choices and to measure the weight that the different factors have on their choices. Often, from the point of view of utility maximization theory, it is assumed that agents measure and compare the available alternatives through a utility function and choose the option that maximizes their utility [45,46]. The theory provides a sound yet simple framework to model travellers' choices and understand the factors influencing their preferences by estimating random utility models [47–49]. Nonetheless, it has been found that maximum utility theory cannot reproduce human behaviour in certain circumstances. Thus, alternative theories have been developed, such as bounded rationality [50,51] and prospect theory [52–54], that try to account for cognitive biases in the decision-making process. Even though these theories provide insights on different aspects of human choice behaviour, none can be considered 'true', and all have some limitations. Here, we assume that travellers are economic agents who maximize their expected utility or follow the principle of least effort [55], a common assumption among mobility models [56,57]. It is based on the idea that passengers try to minimize their travel costs when choosing between using a car or any other transport method.

Travellers choose either of the modes of transportation, minimizing their travel cost. However, they also induce a cost to the rest of the travellers (marginal social costs), especially through travel time, as an extra car on the road implies a reduction of the speed and thus higher travel times for the rest of the users on both car and public transport. Likewise, an extra user in a saturated public transport induces a cost to the rest of the travellers, for example, by longer alighting times and the need for more vehicles and infrastructure that reduce the speed on the roads. In general, there exists a reference scenario from which congestion creates a negative externality on the population. Since each traveller is trying to minimize its travel cost, but at the same time, induce a negative cost to the others, there is a feedback loop between the mode choices of individual travellers and the states of the network.

## 2.2. Travel time as the commuting cost

Travel time has been found to be the strongest predictor of mode choice [36,58]. In New York, for example, a 15 min shorter commuting time corresponds to about 25% higher ridership for a rail service [59]. A survey of car users conducted in Amsterdam showed that when the perceived travel time ratio between car and public transport dropped below 1.6, participants were willing to regularly make the trip by public transport [58]. The effects of congestion on social welfare highlights travel time as the proxy to address individuals' cost [60–64].

While being an important factor influencing travellers' mode choice, travel time is not the only relevant one when promoting a shift from car to public transport [65]. Travel time reliability, for example, can also play an essential role in the mode choice of travellers [33]. A survey conducted in Auckland shows that commuting control and high travel time reliability are key factors in choosing cycling as the mode of transportation [66]. In the case of public transport, headway regularity has a

positive effect on the satisfaction of users [67]. Also, it has been found that the built environment (population density, mixed land use, wide sidewalks, aesthetics and the presence of public transport facilities) promotes walking [68,69], and that providing information to travellers on the social costs associated with the alternatives (travel time or greenhouse gas emissions), can influence their decisions towards a more social route or mode of transportation [70,71]. Other elements, such as limiting motorized vehicles in high-density areas of the cities, also promote the use of public transport [33]. Furthermore, some people might have restrictions on their decision. For instance, a budget restriction might stop many from driving a car; some people do not know how to cycle or drive, or a person with reduced mobility might not consider walking as an option. Let us ignore any restrictions, and so assume that all people could select all commuting methods.

Although not a perfect metric, we assume that travel time is a good proxy of the cost of using any transport mode for simplicity. Therefore, we consider that it is possible to assign a cost of using a car or any other system and that the cost can be measured in travel time, or *commuting minutes*. The rationale behind this choice is that travel time is an increasing function of the number of travellers using a particular mode of transportation. It is a variable that travellers experience immediately after each trip, thus influencing their short-term decisions. Furthermore, the effects of congestion on social welfare highlight travel time as the proxy to address individuals' cost [60–64].

## 2.3. Transport mode as an equilibrium

Equilibrium is an essential concept in transportation studies. It is mainly analysed within the context of traffic assignment (route choice), where trips are distributed to the different roads in the network by assuming the assignment is in equilibrium, in which all travellers choose the route that minimizes their travel cost (travel time) and thus no user can further minimize its travel cost by unilaterally changing routes [45,46,51]. We analyse an equilibrium but not in terms of traffic assignment between different paths, but in terms of mode choice, between cars and other transport systems.

When travellers' choices (cost minimizers) are considered altogether during a period of time, a collective behaviour emerges. Suppose all individuals in a city decide to drive. The chances are that traffic jams and the lack of parking spaces create an enormous cost for everyone. Some people will explore their opportunities and decide to walk, cycle or try public transport instead, reducing their own cost. More users might choose to walk or cycle as they notice that other users experience fewer costs that way, and so more cyclists or public transport users are expected as some replicator dynamics. Yet, they impose now a cost on other walkers, cyclists or public transport users. With more public transport users, queues and delays might become more frequent, and rush hour becomes less comfortable. With more cyclists, bike lanes get more saturated and spaces to leave their bike become scarcer. With more walkers, sidewalks experience congestion. Eventually, an equilibrium is reached, where under the same conditions (say, a regular Monday morning) drivers are not inclined to stop using their car, as they do not minimize their cost by doing it so. Public transport users, walkers or cyclists are also not seduced to change their commuting method either, as their costs have also been minimized. A similar equilibrium is reached if everyone decides to walk or to use public transport one day. Some people might decide to explore their opportunities by driving (perhaps those with the budget), reducing their costs by doing it so, eventually reaching an equilibrium as more people mimic their behaviour.

We can theoretically assume a network to be in equilibrium if every individual is trying to maximize their satisfaction (unknown for us) by means of their (imperfect) perception of the world states and the limitations of their environment. In practice, establishing if a city is in equilibrium is more complex, as our observations about the decision-makers and the choice situations are limited. In idealized scenarios, such as laboratory decision experiments, it has been found that equilibrium is reached after repeated choices of participants [72,73]. However, establishing if an equilibrium is achieved at a city scale requires knowing the weight that the different factors have in the preference of the decision-makers and the states of these factors when people made their choices. Since travel time is a variable that can be measured in a real-world setting (through GPS traces, for example), in practice, it is often assumed that it is the only variable that intervenes in the decisions of travellers. In the context of route choice, for example, it has been found that at an individual level, car drivers do not follow the fastest path, but that the traffic states of the network are close to the theoretical user equilibrium, which considers that travellers are travel time minimizers [74]. Furthermore, even though travellers do not follow the fastest path, there is a threshold for the extra time they are willing to incur by choosing suboptimal routes [75–79].

If we assume that a transport system reaches equilibrium (in the morning peak hour, for example), then the trips' modal share is in equilibrium. Recently, it was estimated that using public transport in São Paulo, Sydney, Stockholm and Amsterdam takes, on average, between 1.4 and 2.6 times longer than driving a car [44], although the ratio might be smaller for short distances. Users are willing to have an extra commuting time for long distances when using the public transport of up to twice the driving time. In turn, the fact that there is a roughly similar travel time ratio suggests that cities reach a transport equilibrium, where some aspects of public transport and active mobility compensate for the fact that they take twice as long for a regular commute. But also, the stable ratio indicates that with extra travel time in a car, say due to the excess number of vehicles, the rush hour or some policies, like reducing the space in cities devoted to cars and reducing parking, people will probably choose to use the public transport instead. Yet, a change in the factors that influence travellers' decisions, or a change in their preferences, will move the stability. This has been observed, for example, in mobility reductions [80], fewer users of public transport [81] and people who now find walking and cycling more attractive as a result of the COVID-19 pandemic [82].

## 2.4. The price of anarchy

A system in equilibrium does not necessarily entail the minimum system total cost (system optimum). When travellers act in their own interest (selfish behaviour) by choosing the mode of transportation that minimizes their cost, the system usually faces some extra costs compared with those obtained if they acted cooperatively. The system's loss of efficiency due to non-cooperative behaviour is called the *Price of Anarchy* [83], defined as a quantitative measure between the worst possible equilibrium and the social optimum. In the context of route choice, selfishly motivated users decrease the network performance [84] creating more traffic. It has been suggested that the loss of efficiency of an urban car network can be up to 33%, meaning that travellers could spend up to one third more of their time in congestion by not behaving cooperatively [85,86].

Assuming that individuals minimize their commuting time and reach a collective traffic equilibrium has given valuable insights in terms of traffic management, including the analysis of some extreme cases and some paradoxical behaviours, where, for example, a new route could increase travel time for all users [87]. Known as 'Braess' paradox' [88], for example, it was shown that, under certain conditions [89,90], when users try to minimize their travel time, due to the side effects of individual actions in the collective traffic, they can incur higher global costs [91].

Based on these results, traffic management strategies to direct traffic flow towards the system optimum have been proposed, such as road use pricing and providing travel information to people with system optimal recommendations [92,93]. However, the price of anarchy and the paradoxical behaviours and traffic management strategies have been formulated in terms of drivers sharing a congested road network, so that it is the traffic assignment between the links that creates these extra social costs. Here, instead of a congested road network, we consider the modal share of a city and the additional cars that citizens use due to selfish behaviours. Thus, the price of selfish drivers is reflected not only in extra commuting time but also in terms of the additional cars and their costly urban impacts.

## 3. Methods

Consider a population of $N$ individuals. To simplify things, we only consider two options: people using a private car or using anything else (called 'public transport users' but includes all non-motorized ways of commuting and public transport). Let $C$ be the number of individuals that drive a car and $T = N - C$ the number of individuals using any other mode of transportation. Assume that travellers are homogeneous, in the sense that the cost they incur by choosing one or the other transportation mode is the same for all travellers. The 'cost' of using the car can be expressed as $C_0 + f_C(C) + g_C(T)$, where $C_0$ is the baseline 'cost' of using a car in the city, $f_C(C)$ are the extra costs that car users put on themselves (such as congestion), and $g_C(T)$ is the extra costs that public transport users put on car users (such as public transport lanes). Let $T_0 + f_T(C) + g_T(T)$ be the baseline and the additional costs that car and public transport users have on people who use public transport (such as traffic jams, queues and delays in the system due to the extra users, respectively). To keep the model simple, assume that $f_C$, $g_C$, $f_T$ and $g_T$ are linear and suppose that all cost units can be measured in commuting time (min), so that $C_0$ is the mean commuting time in the city experienced by car users in free-flow, i.e. in a non-congested city, and the commuting time for car users is

$$\text{Driving cost: } C_0 + \alpha C + \beta T, \tag{3.1}$$

and the commuting time for public transport users is

$$\text{Public transport cost: } T_0 + \gamma C + \delta T, \tag{3.2}$$

where the coefficients $\alpha, \beta, \gamma$ and $\delta \geq 0$ are the marginal social costs, i.e. the extra commuting time induced by an extra car user ($\alpha$) and a public transport user ($\beta$). Similarly, the extra commuting time by public transport induced by an extra car user is $\gamma$, and by an extra public transport user is $\delta$. The commuting time that a driver experience is the addition of three components: a baseline time $C_0$, the extra cost due to other drivers in the city, $\alpha C$, and some extra time that walkers, bikers and public transport users impose on the roads, $\beta T$.

All individuals choose between using a car or the public transport ($N = C + T$) so that the cost for car drivers can be expressed as

$$\text{Driving cost: } C_0 + \beta N + (\alpha - \beta)C, \tag{3.3}$$

and the cost for public transport users is, as a function of the number of car users,

$$\text{Public transport cost: } T_0 + \delta N + (\gamma - \delta)C. \tag{3.4}$$

Assume that the commuting time for car drivers does not decrease with more car drivers (even if there is an extra cost due to the extra public transport users) so that

$$\alpha - \beta \geq 0, \tag{3.5}$$

and similarly for public transport users, so that

$$\delta - \gamma \geq 0. \tag{3.6}$$

The mean cost of commuting for the city for a certain combination of $C$ car users and $N - C$ public transport users, $\mu(C)$, is given by

$$\mu(C) = \frac{(\alpha - \beta - \gamma + \delta)C^2 + (C_0 - T_0 + \beta N + \gamma N - 2\delta N)C + NT_0 + \delta N^2}{N}. \tag{3.7}$$

Notice that assumptions (3.5) and (3.6) imply that $\mu(C)$ is a convex parabola. The mean cost with zero car users is $\mu(0) = T_0 + \delta N$, which is the baseline cost for public transport users ($T_0$) plus the *induced cost* that public transport users inflict among themselves, $\delta N$. The mean cost with only car users is $\mu(N) = C_0 + \alpha N$, again a combination of the baseline cost ($C_0$) and the induced cost $\alpha N$.

## 3.1. A social dynamic

Consider a dynamic in which users, who can be thought of as agents [94–96], choose whether to use their car or an alternative at time $t + 1$ based on their costs and the average costs of everyone else at time $t$, that is, based on their experienced commuting time and the average commuting time. Let $C_t$ be the number of car users at time $t$, and consider that individuals have sequential decisions [97], and compare their commuting times (or costs) with the alternative method. If the commuting times between the two systems are not equal, then a small portion of the more costly method, $\pi > 0$, proportional to the difference in costs, decides to move to the more efficient mode at time $t + 1$, the next day. A discrete dynamic, similar to the replicator equation [98], is given by

$$C_{t+1} = C_t + \pi(T_0 + \delta N + (\gamma - \delta)C_t - C_0 - \beta N - (\alpha - \beta)C_t), \tag{3.8}$$
$$= C_t(1 - \pi(\alpha - \beta - \gamma + \delta)) + \pi(T_0 - C_0 + N(\delta - \beta)). \tag{3.9}$$

Under this dynamic, $C_{t+1} > C_t$ if $C_t < (T_0 - C_0 + N(\delta - \beta))/(\alpha - \beta - \gamma + \delta)$, so that more people decide to use a car if it is less costly than the public transport. Values of $\pi \in (0, 2/(\alpha - \beta - \gamma + \delta))$ guarantee that the dynamics eventually reach an equilibrium and avoid cyclic or extreme behaviours (see appendix A.2). The intuition behind the equilibrium is that a city eventually reaches a point in which drivers are not motivated to use public transport, and if they ever try it, they experience longer commutes and probably go back to driving their cars. The same happens for public transport users, so eventually, the number of drivers and public transport users becomes stable. The dynamic describes a unique Nash equilibrium with non-cooperative self-interested agents [99] and is a frequent technique to analyse selfish routing and its inefficiency [100,101].

There are two extreme cases. If public transport is less costly than using a car, for all values of $C_t$, then all users will end up using it instead of a car. Similarly, if using a car is less costly than the public

transport no matter what the number of users is, then under the dynamics all users will end up using a car. Let us assume that this is not the case, so that there is a value of $C$, say $C^\circ \in [0, N]$, such that the costs of using a car and using the public transport are equal, and that $C_{t+1} > C_t$ if $C_t < C^\circ$ and $C_{t+1} < C_t$ if $C_t > C^\circ$. Therefore, $C^\circ = (T_0 - C_0 + N(\delta - \beta))/(\alpha - \beta - \gamma + \delta)$.

The existence of $C^\circ$ with reasonable values (meaning $C^\circ \in [0, N]$) is guaranteed if the cost function for car users intersects the cost function for the public transport users. In other words, if the city is such that if everyone uses a car, then due to the congestion, public transport is better, but if everyone uses public transport, due to the queues and saturation, then a person could minimize their cost by using a car. There exists a number, $C^\circ$, in which both modes of transportation are in equilibrium, i.e. no person can minimize their own travel time by unilaterally changing the mode of transportation, and the costs of using a car are the same as using public transport.

There are thus, two restrictions

$$\beta \leq \delta + \frac{T_0 - C_0}{N} \quad \text{and} \quad \gamma \leq \alpha - \frac{T_0 - C_0}{N}. \tag{3.10}$$

Under the dynamics (equations (3.8)), there is one fixed point $C^\circ$ and it is a stable attractor node and the only equilibrium of the system, so that regardless of where the city started, eventually they will converge to $C^\circ$ car users and $T^\circ = N - C^\circ$ public transport users. The mean cost of the fixed point is $\mu^\circ = \mu(C^\circ) = C_0 + \beta N + (\alpha - \beta)C^\circ$.

## 3.2. Measuring social inefficiency

The function $\mu(C)$ reaches its minimum value for some $C = C^\star$, given by

$$C^\star = \frac{T_0 - C_0 - \beta N - \gamma N + 2\delta N}{2(\alpha - \beta - \gamma + \delta)}, \tag{3.11}$$

and then increases with more or fewer car users than $C^\star$. The value of $C^\star$ with reasonable values (meaning $C^\star \in [0, N]$) is guaranteed with restrictions given in equation (3.10), meaning that if the costs of using a car and using public transport are equal at some point, then the minimum mean cost is reached for some value $C^\star \in (0, N)$. See appendix A.1.

The minimum mean cost is given by

$$\mu^\star = \mu(C^\star) = T_0 + \delta N - \frac{(C_0 - T_0 + \beta N + \gamma N - 2\delta N)^2}{4N(\alpha - \beta - \gamma + \delta)}. \tag{3.12}$$

For a combination of $C$ car users and $N - C$ public transport users, define their *excess cars* as $\eta(C) = C - C^\star$, assuming that the city has more cars than the optimum number, and zero otherwise (if there are fewer cars than the optimum). The system of equations is symmetric with respect to $C$ and $N - C$, but we focus on the excess number of cars as it is the social costs of the excess vehicles, rather than simply the extra congestion due to more cars, that we are interested in. Consider also the *inefficiency* $\chi(C)$ as the rate between their mean cost and their minimum mean cost, $\mu^\star$. Formally, let

$$\chi(C) = \frac{\mu^\star}{\mu(C)}. \tag{3.13}$$

Notice that $\chi(C) \in [0, 1]$ and values close to one mean a more 'efficient' combination of car users and public transport users and can be interpreted as the level of efficiency that individual decisions have on a collective outcome.

The model depends on four non-negative parameters $\alpha$, $\beta$, $\gamma$ and $\delta$ and two initial costs $C_0$ and $T_0$. Given the restrictions (equation (3.10)), a combination of the four parameters and two initial costs produces a fixed point (the equilibrium) $C^\circ$ and an optimum number of cars $C^\star$, both in $[0, N]$ and thus, the four parameters and two initial costs yield a value of the *social excess cars* as $\eta(C^\circ) = C^\circ - C^\star$, assuming that the equilibrium yields more cars than the optimum value, and a *social inefficiency* $\chi(C^\circ) = \mu^\star/\mu(C^\circ)$ (figure 1). The social inefficiency $\chi(C^\circ)$ is the inverse of the price of anarchy [83,84]. In general, it has been shown that the price of anarchy does not have an upper boundary [93], although, for linear costs (as we have defined here), it is bounded above by 4/3 [84]. Thus, if instead of considering linear costs from car and public transport users, we applied more general functions, $\chi(C^\circ)$ would still be in the interval $[0, 1]$.

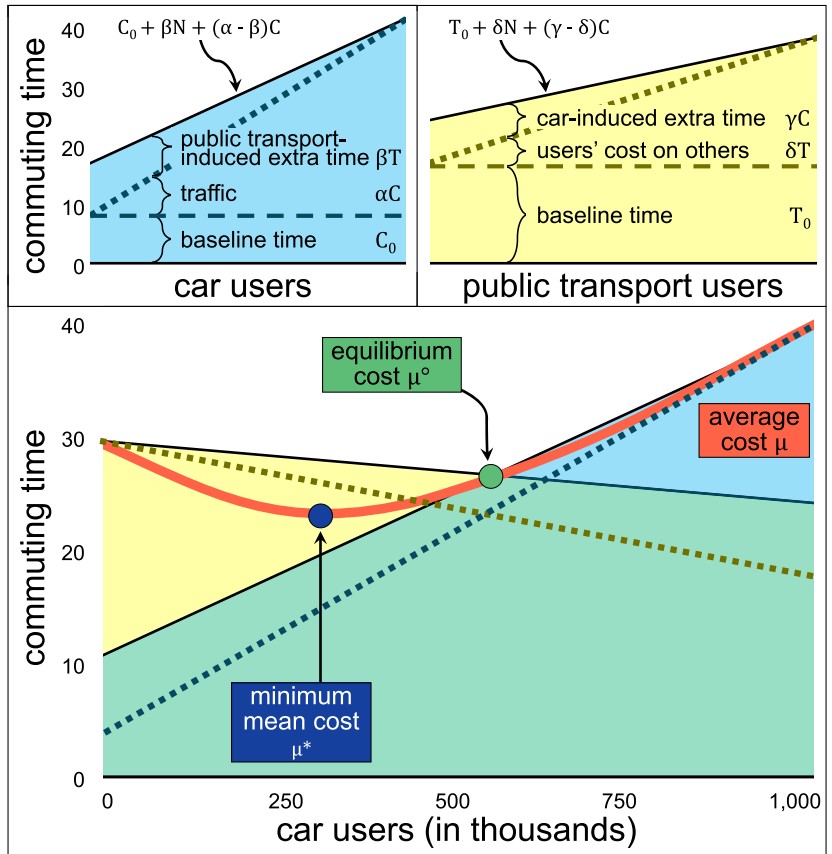

**Figure 1.** Commuting time (vertical axis) as a function of the number of users (horizontal axis). In the top left panel, the commuting time of car users are divided into (i) a baseline cost, (ii) a cost induced by other drivers, which increases with more car users (traffic), and (iii) a cost induced by public transport users, which increases with more public transport users (so it decreases with more car users). The top right panel represents the cost of public transport users, and it is analogous to the car costs. The horizontal axis from the top two panels has opposite directions, as one measures car users and the other public transport users. Both cost models, combined with the same horizontal axis, form the model in the bottom panel. Considering a population of $N = 1\,000\,000$ individuals, the average cost, as a function of the number of car users, is the red parabola. The cost at equilibrium, $\mu^{\circ}$, is observed when both modes' travel cost is the same. However, the minimum attainable cost, $\mu^{\star}$, lower than the equilibrium cost. The system produces $\eta(C^{\circ}) = 200\,000$ extra cars in the city and a social inefficiency of $\chi(C^{\circ}) = 0.9$.

## 4. Results

The parameters' impact is analysed by pairs, considering a population of $N = 1\,000\,000$ individuals and simulating the dynamics under equilibrium (figure 2). For each set of parameters, the social inefficiency $\chi(C^{\circ})$ and the social excess cars $\eta(C^{\circ})$ are reported.

With the same extreme costs, but varying the baseline (left panel of figure 2) the values of $\mu_0$ and $\mu_N$ are fixed, so the users experience the same cost $\mu_0 = T_0 + \delta N = 45$ min if there are no car users, and $\mu_N = C_0 + \alpha N = 60$ min if everyone uses a car. Varying $C_0$ and $T_0$ is equivalent (in this case with fixed extremes) to varying the values of $\alpha$ and $\delta$. The extreme values of the average cost, $\mu(C)$, are fixed, and the gradient of the public transport and the car costs vary according to the parameters. Results show that the social system is more efficient with higher values of $C_0$ (so that the costs of using a car are not the result of the induced traffic from other users but a costly baseline), and the city also has fewer excess cars. Also, the social system is more efficient with smaller values of $T_0$, meaning that most of the cost of using public transport is either induced by other public transport users or by car drivers. Notice that the different direction between $C_0$ and $T_0$ to obtain a more efficient social system is obtained because of the values of $\mu_0 = 45$ and $\mu_N = 60$. With the opposite values in the extremes ($\mu_0 = 60$ and $\mu_N = 45$), the opposite results are obtained due to symmetry.

With fixed values of $\mu_0 = 45$ and $\mu_N = 60$ (middle panel of figure 2) but varying the costs that private cars have on public transport users ($\gamma$) and the cost that public transport users have on cars ($\beta$) results

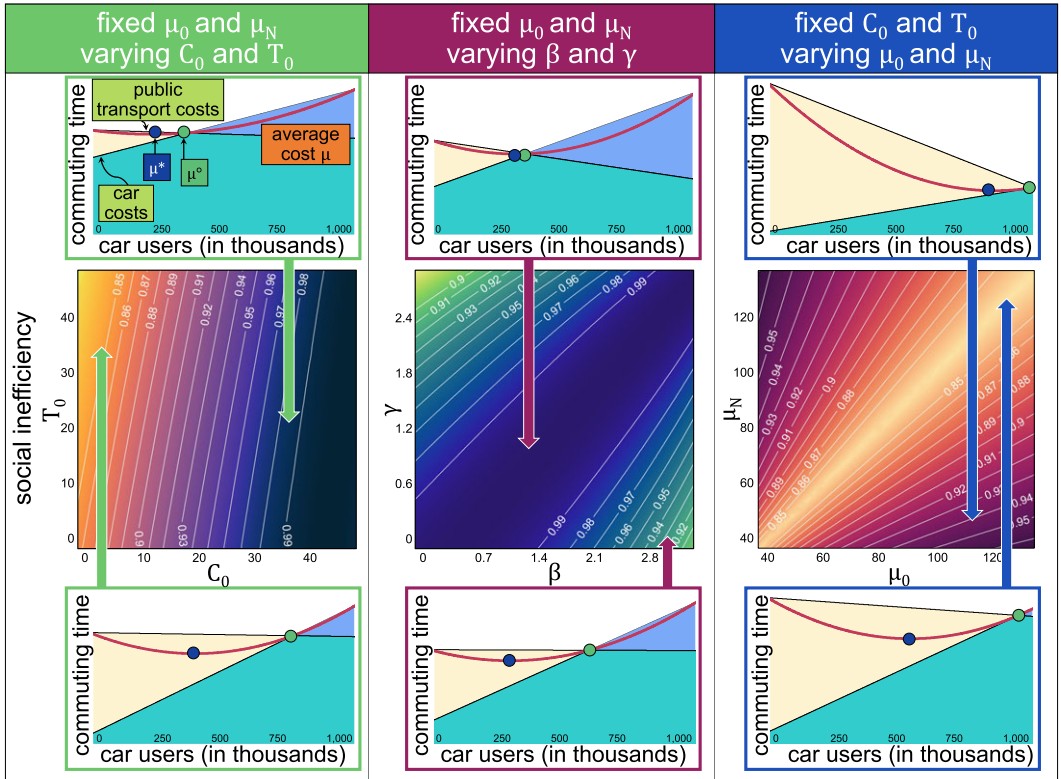

**Figure 2.** Impact of the model's parameters in terms of the social inefficiency $\chi(C^\circ)$. The left panel shows a system where the extreme average costs are fixed, but varying baseline costs $C_0$ and $T_0$. Although $\chi(C^\circ) \approx 1$ in the upper-left panel, it still produces $\eta(C^\circ) = 125\,000$ extra cars. The middle panel shows a system with fixed extreme average costs but varying the induced costs of car users on public transport users $\gamma$ and of public transport users on car drivers $\beta$. The right panel shows the impact of changing the extreme average costs, where again, the upper-right panel has $\chi(C^\circ) \approx 1$ but produces $\eta(C^\circ) = 125\,000$ extra cars. In all three panels, the lighter regions have more social inefficiency, which is observed when the equilibrium cost $\mu^\circ$ (green circle) is farther from the minimum average cost $\mu^\star$ (blue circle) and the three bottom figures produce between $\eta(C^\circ) = 350\,000$ and $400\,000$ additional cars.

show that the system is more efficient if the induced costs are of similar magnitude (that is, $\beta$ and $\gamma$ have similar values). The system is more inefficient and produces a higher number of excess cars when values are not of similar magnitude, meaning that if the cost that 1000 drivers have on public transport users is higher than the cost that 1000 public transport users have on drivers, the system is less efficient. In this case, the excess cars in the city can increase drastically.

Finally, varying the values of $\mu_0$ and $\mu_N$ and keeping the baseline costs $C_0$ and $T_0$ fixed (right panel of figure 2) means that $\alpha$ and $\delta$ vary accordingly, and most of the costs are induced by others. Results show that with similar values of $\mu_0$ and $\mu_N$ the social system is less efficient and produces a higher number of excess cars, whereas having $\mu_0$ and $\mu_N$ with considerably different values gives a more efficient system.

In general, the social system is less efficient if the baseline costs of driving a car are low, and there is a high induced cost from other drivers—congestion—and at the same time, there is a high cost for public transport users (bottom left panel of figure 2). Also, the system is less efficient if car users put a high cost on public transport users (bottom centre) or if public transport users do not observe a lower cost with more car users and less public transport users (bottom right). The average commuting time can increase considerably (efficiency lower than $\chi(C^\circ) \approx 0.8$), and there could be more than $\eta(C^\circ) = 400\,000$ extra car users than the social optimum if such conditions are observed. The case in the bottom left panel shows an equilibrium with a large average commuting time, nearly 20% more than the optimum and shows that society has 350 000 extra car drivers than the best-case scenario, as there is a high cost for public transport users and also, congestion creates a high cost for drivers. Probably scenarios like the ones in the bottom of figure 2 are observed in cities such as Detroit or Atlanta, where more than 90% of the individuals decide to drive a car. As a result, high congestion is an imposed cost from society.

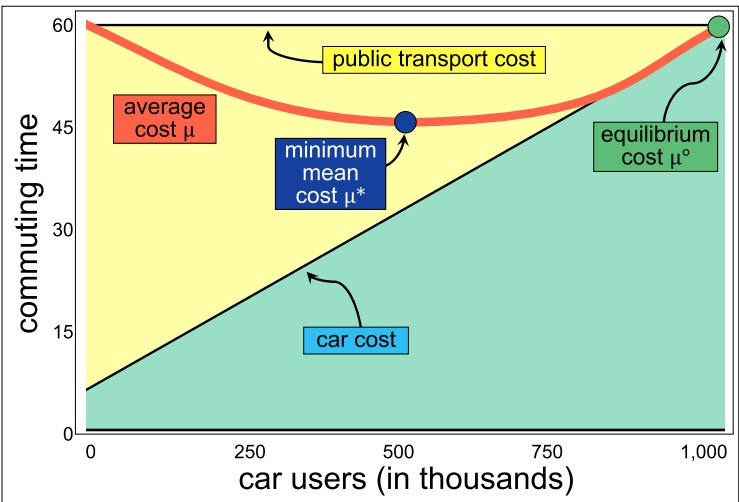

**Figure 3.** Model of social costs (vertical axis) as a function of the number of car users (horizontal axis) considering a population of $N = 1\,000\,000$ individuals, $T_0 = 60$ min, $\alpha = 55/N$ and $\beta = \gamma = \delta = 0$, so that drivers have an extra cost of $55C/N$ min due to congestion and a value of $C_0 = 5$ min. Under the scenario, $C^\circ = N$, so all individuals are inclined to use their car, with a mean cost of $\mu^\circ = 60$, most of it the result of induced traffic. With $C^\star = 500\,000$ car users, $\mu^\star = 45 + C_0/4$, so that the system inefficiency is $(45 + C_0/4)/60 = 0.75 + C_0/240 \approx 0.77$.

## 4.1. The extreme case: a paradox

We explore the limit case, where parameters are adjusted such that all the restrictions are met, and the result is the highest possible social inefficiency. Consider the case in which most of the costs of using a car are induced by other drivers, so small baseline cost $C_0$ and a large value of $\alpha - \beta$, and the costs of using the public transport are high but independent of the number of users, so a large baseline cost $T_0$ and a small value of $\delta - \gamma$. In this limit case, only with full saturation $C = N$ the costs of using a car are the same as using the public transport (figure 3, where the values of the parameters are $C_0 = 5$ min, $T_0 = 60$ min, $\alpha = 55/N$ and $\beta = \gamma = \delta = 0$).

If $k$ people decide to use public transport, their cost would be 60 min, whereas the rest of the users' cost is $60 - 55k/N < 60$. Thus, under the limit case, driving a car reduces their commuting time, and they might be inclined to use their car instead as a result. However, when all individuals decide to drive a car, their commuting time is 60 min, the same cost as using public transport, and the worst average time. All citizens try to minimize their own cost, and they do it by driving their car. Yet, collectively they converge to the overall worst-case scenario, where the average commuting time is maximum. Further, the excess number of cars in this scenario is $\eta(C^\circ) = 500\,000$, meaning that the city has half a million additional vehicles than it should. All people decide individually and minimize their own costs, but collectively they maximize the average cost and the cost for everyone.

The paradoxical result, in the case of selfish routing, has been explored before [93,102] and it is the setting used for measuring how bad is selfish routing with linear costs [84] (with an upper bound in the price of anarchy of 4/3). There are other paradoxes observed in mobility studies. For instance, Braess' paradox [88] shows that (given the right set of parameters) removing an avenue from the city could improve traffic assignment among routes. Here, we observe that given the right set of parameters as well, people trying to minimize their own commuting time is not just inefficient, but could also give as a result the worst overall scenario with nearly twice as many cars as the city optimum. Beyond that, the observed paradox also shows that the city would use one million vehicles (with one million drivers), considering the social cost of everyone driving.

Collectively, an individual decision-making process might lead to the worst case, described by Anthony Downs as 'the most serious environmental problem', since, as opposed to poverty or obesity, traffic congestion and its by-products are suffered by everyone [103]. Thus, the challenge with selfish routing is how to make the system more efficient [101] and with fewer cars. Some proposed techniques to increase efficiency are to route some users centrally [93] or through congestion charges [103], which differentiate public and private transportation increasing the private costs to stimulate the public [104–107]. In fact, a random decision-making process for all individuals would be better than all acting selfishly, for instance, by flipping a coin. Each person flips a coin and uses a car or public

transport according to the outcome. We expect that half of the people would use the public transport (with a cost of 60 min, the same as if everyone uses a car), and half of the people are expected to drive a car (and observe a commuting time of 32.5 min), meaning that they save nearly half an hour and the city would require half the number of cars. People flipping a coin (or another system, such as taking the last digit of the person's ID) is a better social coordination system than all people acting selfishly, trying to minimize their own commuting time.

# 5. Conclusion and discussion

The rapid growth of urban population, land-use policies and car-oriented infrastructure that dominated the twentieth century derived in sprawling low-dense suburban areas, increasing the commuting distance at the expense of active modes of transportation (walking and cycling), and making it economically costly to introduce efficient public transport with high frequency and reachable at short walking distance [108,109]. The car became the preferred mode of transportation for many city-dwellers, increasing the number of motorized trips and, as a consequence, augmenting the congestion and air pollution. Moreover, the large amount of public resources put into car infrastructure (roads, freeways, parking lots) and the bailouts paid to the car industry for the 2008 global financial crisis are subsidies that hide the real economic costs of car use, not to mention the unethical industry efforts to hide the negative environmental and health impacts of car use [110–112]. In this context, it would be difficult to claim that our choices are rational since information and environmental constraints impact our choices. Providing citizens with more travel options, local shops and services with mixed land use can decrease car use [113,114]. Furthermore, information about their choices' social costs can influence their travel decisions towards more socially efficient alternatives [71,92]. Still, changing mobility behaviours is challenging and might require interventions such as congestion charges, dynamic tolling, parking bans and removing free parking to reduce single-occupancy vehicle commutes, increase public transport use and promote active mobility [115].

Mobility is a crucial aspect to consider both for urban studies and for sustainability. Producing cars takes 4% of the total carbon dioxide emissions, but there are all types of other costs related to motorized mobility. These include direct costs, such as the petrol or electricity they consume, infrastructure and congestion itself, and indirect ones, including road insecurity, the (un)active mobility, the space devoted to cars in cities and others. Reducing the use of private vehicles in a city plays a fundamental role in the already existing car-dependent cities, such as Detroit and Atlanta, but also where it is still desirable to use a car but budget restrictions limit growth, as in Mexico City or in cities where the population is estimated to increase to 60, 80 or even 100 million inhabitants, such as Lagos and Dhaka [116]. Reducing car-dependence and promoting non-motorized mobility, such as walking and cycling, and promoting the use of public transport for longer distances, forms the basis of modern urban planning.

From economics, there is an extensive tradition in welfare theory where the tragedy of the commons and the concept of externality have a relevant role. An externality is any positive or negative effect of a decision, such as driving a car, experienced by other individuals than the drivers. Therefore, it needs to be integrated into their decision process; usually, the government enforces the inclusion of the externality into individuals' decision process through law or similar, because they would not do it by themselves [117]. Examples of negative externalities and its attempt to correct them are environmental taxes on polluting industries or cash transfers in the education or health sector. In these examples, the purely individual behaviour creates an inferior situation compared with the potential maximum social welfare [118]. Economics acknowledges congestion as a negative externality that needs to be corrected to maximize social welfare. Solutions in this direction have implemented pricing congestion through road toll as an effective policy [62,63,119,120].

The model emphasizes the need for collective behaviour using a parsimonious approach with minimal assumptions that are feasible [121,122]. Even though each journey is different, we are trying to observe the emergence of a collective decision-making process to generalize from such individual aspects to a general, city-level observation of public transport use. Similarly, from an economics point of view, lower prices of a particular good are assumed to increase demand, although many individual aspects are ignored, favouring a collective observation. With a simple model to account for the costs of commuting induced by other users measured in time and, assuming a dynamic in which individuals chose the least costly method, we grouped the 'desirable' modes of transportation into a single group and measured the excess number of car users that would be observed. Under different scenarios, we observe a very inefficient social system. There could be more than 40% extra drivers in

the city (and 40% extra cars) and commute 20% or more extra time than needed under the best social scenario. They pay a high price for congestion and the rush hour. This is an emergent collective pattern, where all individuals try to minimize their own travel time, but that might result in very inefficient social scenarios. The limit case is actually a paradoxical result. All individuals drive their own car, and all minimize their commuting time, but collectively arrive at the worst-case scenario, in which the average cost is the highest.

The model could be considered to be simplistic. First of all, considering all individuals as being homogeneous, with equal cost functions and no barriers to choosing a transportation mode. Secondly, putting private cars in one group and the rest of the modes in the other ignores the differences in costs and travel times between walking or taking the bus or the metro. Thirdly, the replicator's dynamic simplifies how commuting methods are chosen, as there are many more aspects to take into account, including budget restrictions or whether the person knows how to drive. And finally, it is assumed that cities eventually reach an equilibrium in terms of the collective transportation mode. The model could be improved considering individual differences, social elements or city restrictions and spatial aspects of the journey. Still, while simple, our results have some direct policy implications, which are analysed in terms of what a city planner can do to improve their city mobility.

Collectively, we reach a critical point with extra cars and a higher travel time due to individual actions. As such, that is the paradox we encountered. All individuals try to minimize their travel time but collectively reach the worst-case scenario and a Nash equilibrium, where no person is motivated to alter their travel choice. Although it presents many challenges at a practical level (due to its implementation and the unpopular cost on those banned from driving), it shows that a collective decision-making process might be highly inefficient. Under such conditions, a better decision-making process (at a collective level) allows only some drivers in the city randomly. Some cities, including Mexico City, Beijing, Athens, Quito and Bogotá, have introduced programmes that ban drivers from using their vehicles on some days of the week [123] although the impact of such programmes tends to be limited and lasting only a few months [124]. In Mexico City, for example, a policy implemented in 1989 banned vehicles based on their licence plate, but the policy led to an increase in the total number of cars [123]. In the long run, the policy encouraged drivers to buy additional (frequently cheaper, older and hence more polluting) vehicles to circumvent the restrictions [125]. As long as driving in a city is more secure, more comfortable and faster than using public transport, cycling or walking, people will eventually be seduced by a car, and many will ultimately drive. Convincing drivers of avoiding single-occupancy vehicle commutes, using public transport, cycling and walking cannot rely on a single policy but on increasing journey options, accessibility and time efficiency. A city needs fewer and shorter journeys, accessibility so that people can walk, cycling lanes and more public transport, so that the best transport mode for everyone is not to use a car.

## 5.1. City improvements: untangling costs of walking, cycling and public transport

The costs of the non-motorized modes of transportation were grouped into three components. Firstly, the baseline cost, which is the expected commuting time if a person was alone in the city and decides to use public transport or to walk. Secondly, the induced cost by other users, such as queues, delays, passenger drop-off and pick-up, busy trains, stairs, streets or cycle lanes. Thirdly, the cost that car drivers put on public transport users,

Each one of the three types of cost can be reduced with different interventions, and each type of intervention and cost reduction has a different impact. For example, the baseline cost requires more infrastructure (a new bus line or a new metro line), but constructing a new metro line can be extremely expensive and our results show that it could be inefficient if reductions in the baseline cost get compensated with an increased—induced—cost from the other public transport users. Take, for instance, the scenario at the bottom left of figure 2. Reducing the baseline cost could decrease the average cost and increase social efficiency just by a small amount (reducing the number of cars and the commuting time by a slight amount as well). Under that scenario, a cheaper and more efficient policy would be to increase the induced costs that public transport users have on drivers (the value of $\beta$). Pedestrian crossings or spaces for bicycle parking could reduce the number of cars and increase the social efficiency *without* increasing the average social cost (as some users will decide to use the public transport instead).

Similarly, if the costs of using public transport increase considerably when the system has more users (for instance, the scenario at the upper right of figure 2), the public transport system suffers from high induced costs from other public transport users (high value of $\delta$). Long queues at the entrance, an

inefficient ticketing system, or congested sidewalks could be the system's norm. Perhaps, instead of more infrastructure, costs could be reduced by redesigning the entrance to the metro or the ticketing system and increasing the frequency of buses and trains.

Finally, the public transport system suffers from a high induced cost from car drivers (a high value of $\gamma$) with vehicles invading the pedestrian crossings, cars using public transport lanes, the construction of pedestrian bridges and other scenarios. Under such conditions, the system could be more efficient and with fewer cars, if priority is given to pedestrians and cyclists when designing streets.

## 5.2. City improvements: untangling costs of driving

Reducing the number of car users could be achieved if their cost is increased, either the baseline cost, the induced cost of other drivers or the cost that public transport users put on drivers. Increasing the baseline cost can be achieved by reducing the speed limit on highways and residential neighbourhoods, traffic lights designed to give priority to pedestrians, reducing the spaces in parking lots and limiting the number of cars allowed to park in the streets or with cash-only parking meters.

Increasing the induced costs that car users put on themselves and that public transport users put on drivers could be achieved with some interventions, by reducing the space devoted to cars, with more public transport lanes, tramways, wider sidewalks and pedestrian roads, for example. Furthermore, in cases in which the cost of public transport is not highly sensitive to its number of users, increasing the baseline cost, or the induced costs of car drivers, as is the case of the three bottom sections of figure 2, the average cost is not increased, and it is obtained with fewer cars in a city.

Data accessibility. All information required to attempt a replication of our work is available in the manuscript and no additional data, code or other material is required.

Authors' contributions. R.P.C. designed the study. H.G.R. and M.Q.D. analysed the results. All authors wrote the manuscript.

Competing interests. We declare we have no competing interests.

Funding. This article was completed with support from the PEAK Urban programme, funded by UKRI's Global Challenge Research Fund, grant no. ES/P011055/1.

Acknowledgements. We thank the anonymous reviewers for their insightful comments and suggestions.

# Appendix A. Supplementary information

## A.1. Existence of $C^{\star} \in (0, N)$

The value of $C^{\star} > 0$ if

$$C^{\star} = \frac{T_0 - C_0 - \beta N - \gamma N + 2\delta N}{2(\alpha - \beta - \gamma + \delta)} > 0,$$

which is satisfied if

$$\beta < \frac{T_0 - C_0}{N} + 2\delta - \gamma,$$

but $\beta < \delta + (T_0 - C_0)/N$ according to inequalities (3.10) and $\delta - \gamma > 0$ according to (3.4).

Similarly

$$C^{\star} = \frac{T_0 - C_0 - \beta N - \gamma N + 2\delta N}{2(\alpha - \beta - \gamma + \delta)} < N,$$

if

$$\gamma < 2\alpha - \beta - \frac{T_0 - C_0}{N},$$

but

$$\gamma < \alpha - \frac{T_0 - C_0}{N}$$

according to inequalities (3.10) and $\alpha > \beta$.

Thus, $C^{\star}$ exists and with reasonable values $C^{\star} \in (0, N)$ if there is a value of $C \in (0, N)$ such that the costs of both methods are equal.

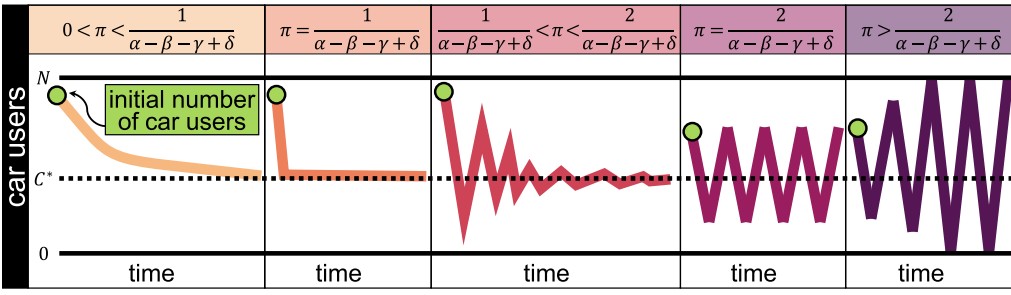

**Figure 4.** Different values of $\pi$ give different dynamics. The first panel shows a simple (slow) attractor to $C^\circ$, and as the values of $\pi$ increase, we get either a one-step dynamic, a damped oscillation, a cyclic oscillation or alternating between 0 and $N$ car users.

## A.2. Existence of a value of $\pi > 0$ to avoid cyclic behaviour

The dynamics from equation (3.8) depend on the parameter $\pi > 0$. Note that

$$C_{t+1} = C_t(1 - \pi(\alpha - \beta - \gamma + \delta)) + \pi(T_0 - C_0 + N(\delta - \beta)), \qquad (A\,1)$$

and assume that extreme values are not reached (so that neither $C_t = 0$ nor $C_t = N$ are observed). It means that $C_{t+2} = C_t$ if $\pi = 2/(\alpha - \beta - \gamma + \delta)$, for which a cyclic dynamic is obtained and with smaller values of $\pi$, each step gets closer to $C^\circ$.

Also, note that if $\pi = 1/(\alpha - \beta - \gamma + \delta)$, then we get that, for any value of $C_t$, then $C_{t+1} = C^\circ$, meaning that the dynamics reaches its stationary point in the first step. Therefore, a summary of the dynamics according to different values of $\pi$ is:

— for values of $\pi \in (0, 1/(\alpha - \beta - \gamma + \delta)$ we obtain a slow approximation to values of $C^\circ$;
— with $\pi = 1/(\alpha - \beta - \gamma + \delta)$, we get the stationary point after one step;
— for $\pi \in (1/(\alpha - \beta - \gamma + \delta), 2/(\alpha - \beta - \gamma + \delta))$ we get a damped oscillation around $C^\circ$, meaning slowly alternating above and below values of $C^\circ$;
— for $\pi = 2/(\alpha - \beta - \gamma + \delta)$ a cyclic behaviour is obtained around $C^\circ$; and
— for $\pi > 2/(\alpha - \beta - \gamma + \delta)$ the dynamic eventually reaches the extreme values and the dynamic simply alternates from $C_t = 0$ to $C_{t+1} = N$.

With values of $\pi < 2/(\alpha - \beta - \gamma + \delta)$, the number of car users under the social dynamics described, eventually reach $C^\circ$ (figure 4).

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
