## [Peer Review File · Royal Society Open Science]

Review History

RSOS-201808.R0 (Original submission)

Review form: Reviewer 1

Is the manuscript scientifically sound in its present form?

Yes

Are the interpretations and conclusions justified by the results?

Yes

Is the language acceptable?

Yes

Do you have any ethical concerns with this paper?

No

Have you any concerns about statistical analyses in this paper?

No

Recommendation?

Accept with minor revision (please list in comments)

Comments to the Author(s)

My background is public health medicine, so I come to transport issues from a different position than the authors of this paper. In my view, the model of transport choices that is applied here leaves out many important factors that affect opportunities and decisions. But the findings are of interest, and they seem to align well with recommendations made on other grounds (eg trials of interventions, analysis of the policy landscape, case studies). The major points, as I see them, are the need for restraints on car use as well as incentives for other modes, that service quality (including comfort, security and attractiveness) are key factors in PT, and that the present transport arrangements are full of perversity, such as the bad results for everyone of increasing individual amenity (supposedly) by building more roads and producing more cars.

My suggestions for minor changes, that I think would improve the paper are:

- acknowledge that it is not just travel time that looms large in every commuter's mind, but the variability in travel time. And in particular how consistent and predictable travel times are. See this paper ('Why are cyclists the happiest commuters?' JTH 2019) and the finding that people who switched to e-bikes were particularly satisfied by the certainty of travel time with this mode.
- I think 'selfish behaviour' is a poor choice of language. I understand I think what is meant here (choices that disregard effects on others). However, in reality, travel choices are conditioned by so many factors, outside the individual's control. Low incomes mean affordable housing for many sits on the far edge of cities, casualised work practices mean commuting is unpredictable and routes change frequently, gender roles bring, for women typically, family responsibilities that strongly influence travel decisions.
- In general, I suggest the paper should acknowledge more prominently the structural and environmental factors that place severe boundaries around 'rational' personal choices. All of us are at the whim, it feels, of the industries that promote cars heavily, lobby for permissive policies and regulation, and are invested heavily in wide-spread urban form and the infrastructure that is in place. Just as an example, and I know this is not directly relevant to the subject of the paper, but it is a powerful story about whose choices matter most, take a look at 'High and Mighty. SUVs: the world's most dangerous vehicles and how they got that way' by Keith Bradsher.
- travellers are economic agents who maximise their expected utility: this is a convenient assumption for the modelling, but is such a flimsy and abbreviated picture of how the world works, I think it needs some mighty strong caveats associated with it. The mobilities literature is packed with studies (including some great papers from the TSU at Oxford!), pointing out the role of experiences such as nature contact, social interactions, physical activity, the sense of agency that accompanies personal effort, the 'nimbleness' of certain forms of travel, and as I have mentioned already, reliability of arrival time. The status associated with certain forms of travel (this is mentioned I think in the paper - and is certainly a factor in places like China where driving really is perverse in the presence of an efficient metro) and the ways in which travel asserts and promotes identities (eg fixie riders in big cities like London - madness, but it's a powerful statement of who you are; drivers of double cab utes likewise).
- I am intrigued by the finding that random allocation of travel mode might produce better results overall than relying on individuals to make good decisions. Does this have any practical applications? Car-free days based on the last digit on the registration plate? Free PT for those born in a particular month of the year?

Review form: Reviewer 2

Is the manuscript scientifically sound in its present form?

Yes

Are the interpretations and conclusions justified by the results?

Yes

Is the language acceptable?

No

Do you have any ethical concerns with this paper?

No

Have you any concerns about statistical analyses in this paper?

No

Recommendation?

Major revision is needed (please make suggestions in comments)

Comments to the Author(s)

The paper deals with a problem of city planning regarding mode of transportations and what modes can lead to better commuting times. The paper defines an equilibrium point but I'm not certain the assumptions are realistic enough for the paper to be very useful in practice.

The issue with the use of motorised means of transportation is not a uniform issue. Clearly the USA has such issues but the same is not true in many European cities. In fact there is evidence of a huge increase in the use of no-motorised modes of transportation (bike shares) in New York, Beijing, Philadelphia (with numbers ranging from 70% to 150% increases). It would be nice if the authors could contrast their results with what is happening in real cities. The analytical analysis could be contextualised with real data. Do cities reach this equilibrium cost?

I found Figure 1 quite hard to understand. For instance, why does the dashed line for public transportation cost with no car change as a function of car users (if there are no cars)?

The paper is an interesting exercise but without evidence of such modelling in real cities, the paper becomes less relevant. Sure it is an interesting modelling but how can we know this model fits or can explain what we see in real cities? At least a simple indication would be needed. I suggest a case study using the data for a couple of cities with different scenarios of public transportation and number of cars (e.g. Miami vs. London)

MINOR COMMENTS

- The literature review is short but perhaps adequate but it misses important works such as the recent works on mobility models.
- There are many grammatical mistakes and typos in the document. It should be carefully proof-read before submitting it again.

Decision letter (RSOS-201808.R0)

Dear Mr Prieto Curiel

The Editors assigned to your paper RSOS-201808 "Modelling traffic and the extra cars in a city as a collective behaviour" have now received comments from reviewers and would like you to revise the paper in accordance with the reviewer comments and any comments from the Editors. Please note this decision does not guarantee eventual acceptance.

Please submit your revised manuscript and required files (see below) no later than 21 days from today's (ie 02-Mar-2021) date. Note: the ScholarOne system will 'lock' if submission of the revision is attempted 21 or more days after the deadline. If you do not think you will be able to meet this deadline please contact the editorial office immediately.

on behalf of Professor Matjaz Perc (Associate Editor) and Mark Chaplain (Subject Editor)
openscience@royalsociety.org

Reviewer comments to Author:

Reviewer: 1

Comments to the Author(s)

My background is public health medicine, so I come to transport issues from a different position than the authors of this paper. In my view, the model of transport choices that is applied here

leaves out many important factors that affect opportunities and decisions. But the findings are of interest, and they seem to align well with recommendations made on other grounds (eg trials of interventions, analysis of the policy landscape, case studies). The major points, as I see them, are the need for restraints on car use as well as incentives for other modes, that service quality (including comfort, security and attractiveness) are key factors in PT, and that the present transport arrangements are full of perversity, such as the bad results for everyone of increasing individual amenity (supposedly) by building more roads and producing more cars.

My suggestions for minor changes, that I think would improve the paper are:

- acknowledge that it is not just travel time that looms large in every commuter's mind, but the variability in travel time. And in particular how consistent and predictable travel times are. See this paper ('Why are cyclists the happiest commuters?' JTH 2019) and the finding that people who switched to e-bikes were particularly satisfied by the certainty of travel time with this mode.

- I think 'selfish behaviour' is a poor choice of language. I understand I think what is meant here (choices that disregard effects on others). However, in reality, travel choices are conditioned by so many factors, outside the individual's control. Low incomes mean affordable housing for many sits on the far edge of cities, casualised work practices mean commuting is unpredictable and routes change frequently, gender roles bring, for women typically, family responsibilities that strongly influence travel decisions.

- In general, I suggest the paper should acknowledge more prominently the structural and environmental factors that place severe boundaries around 'rational' personal choices. All of us are at the whim, it feels, of the industries that promote cars heavily, lobby for permissive policies and regulation, and are invested heavily in wide-spread urban form and the infrastructure that is in place. Just as an example, and I know this is not directly relevant to the subject of the paper, but it is a powerful story about whose choices matter most, take a look at 'High and Mighty. SUVs: the world's most dangerous vehicles and how they got that way' by Keith Bradsher.

- travellers are economic agents who maximise their expected utility: this is a convenient assumption for the modelling, but is such a flimsy and abbreviated picture of how the world works, I think it needs some mighty strong caveats associated with it. The mobilities literature is packed with studies (including some great papers from the TSU at Oxford!), pointing out the role of experiences such as nature contact, social interactions, physical activity, the sense of agency that accompanies personal effort, the 'nimbleness' of certain forms of travel, and as I have mentioned already, reliability of arrival time. The status associated with certain forms of travel (this is mentioned I think in the paper - and is certainly a factor in places like China where driving really is perverse in the presence of an efficient metro) and the ways in which travel asserts and promotes identities (eg fixie riders in big cities like London - madness, but it's a powerful statement of who you are; drivers of double cab utes likewise).

- I am intrigued by the finding that random allocation of travel mode might produce better results overall than relying on individuals to make good decisions. Does this have any practical applications? Car-free days based on the last digit on the registration plate? Free PT for those born in a particular month of the year?

Reviewer: 2

Comments to the Author(s)

The paper deals with a problem of city planning regarding mode of transportations and what modes can lead to better commuting times. The paper defines an equilibrium point but I'm not certain the assumptions are realistic enough for the paper to be very useful in practice.

The issue with the use of motorised means of transportation is not a uniform issue. Clearly the USA has such issues but the same is not true in many European cities. In fact there is evidence of a huge increase in the use of non-motorised modes of transportation (bike shares) in New York, Beijing, Philadelphia (with numbers ranging from 70% to 150% increases). It would be nice if the

authors could contrast their results with what is happening in real cities. The analytical analysis could be contextualised with real data. Do cities reach this equilibrium cost?

I found Figure 1 quite hard to understand. For instance, why does the dashed line for public transportation cost with no car change as a function of car users (if there are no cars)?

The paper is an interesting exercise but without evidence of such modelling in real cities, the paper becomes less relevant. Sure it is an interesting modelling but how can we know this model fits or can explain what we see in real cities? At least a simple indication would be needed. I suggest a case study using the data for a couple of cities with different scenarios of public transportation and number of cars (e.g. Miami vs. London)

MINOR COMMENTS

- The literature review is short but perhaps adequate but it misses important works such as the recent works on mobility models.
- There are many grammatical mistakes and typos in the document. It should be carefully proof-read before submitting it again.

===PREPARING YOUR MANUSCRIPT===

===PREPARING YOUR REVISION IN SCHOLARONE===

To revise your manuscript, log into <https://mc.manuscriptcentral.com/rsos> and enter your Author Centre - this may be accessed by clicking on "Author" in the dark toolbar at the top of the

page (just below the journal name). You will find your manuscript listed under "Manuscripts with Decisions". Under "Actions", click on "Create a Revision".

Author's Response to Decision Letter for (RSOS-201808.R0)

See Appendix A.

RSOS-201808.R1 (Revision)

Review form: Reviewer 1

Is the manuscript scientifically sound in its present form?

Yes

Are the interpretations and conclusions justified by the results?

Yes

Is the language acceptable?

Yes

Do you have any ethical concerns with this paper?

No

Have you any concerns about statistical analyses in this paper?

No

Recommendation?

Accept as is

Comments to the Author(s)

Thank you for the chance to read this paper again. My questions about the first version, and suggestions of possible improvements, have been treated thoughtfully, and the paper is stronger in its present form. I think it warrants publication as is.

Review form: Reviewer 2

Is the manuscript scientifically sound in its present form?

Yes

Are the interpretations and conclusions justified by the results?

Yes

Is the language acceptable?

Yes

Do you have any ethical concerns with this paper?

No

Have you any concerns about statistical analyses in this paper?

No

Recommendation?

Accept with minor revision (please list in comments)

Comments to the Author(s)

The paper is at a much better state than before and I have just a couple of comments still

- Figure 1 continues to be confusing IMO. I understand it may not be possible to change it but could it be broken down into two parts? I think the authors are trying to convey too much in one figure

- I'm still slightly disappointed about the applicability of the results in city planning, city modelling, etc. A stronger discussion session where the authors address the "so what" would be useful. The authors mention in the response that cities appear to reach equilibrium, so can I reverse the question? Are there cities that are not in equilibrium? Can you give an intuition as to whether this non-equilibrium is unstable and that equilibrium must be attained?

Decision letter (RSOS-201808.R1)

Dear Mr Prieto Curiel

On behalf of the Editors, we are pleased to inform you that your Manuscript RSOS-201808.R1 "A paradox of traffic and extra cars in a city as a collective behaviour" has been accepted for publication in Royal Society Open Science subject to minor revision in accordance with the referees' reports. Please find the referees' comments along with any feedback from the Editors below my signature.

Please submit your revised manuscript and required files (see below) no later than 7 days from today's (ie 12-May-2021) date. Note: the ScholarOne system will 'lock' if submission of the revision is attempted 7 or more days after the deadline. If you do not think you will be able to meet this deadline please contact the editorial office immediately.

Kind regards,
Anita Kristiansen

Editorial Coordinator

on behalf of Mark Chaplain (Subject Editor)
 openscience@royalsociety.org

Reviewer comments to Author:

Reviewer: 1

Comments to the Author(s)

Thank you for the chance to read this paper again. My questions about the first version, and suggestions of possible improvements, have been treated thoughtfully, and the paper is stronger in its present form. I think it warrants publication as is.

Reviewer: 2

Comments to the Author(s)

The paper is at a much better state than before and I have just a couple of comments still
 - Figure 1 continues to be confusing IMO. I understand it may not be possible to change it but could it be broken down into two parts? I think the authors are trying to convey too much in one figure

- I'm still slightly disappointed about the applicability of the results in city planning, city modelling, etc. A stronger discussion session where the authors address the "so what" would be useful. The authors mention in the response that cities appear to reach equilibrium, so can I reverse the question? Are there cities that are not in equilibrium? Can you give an intuition as to whether this non-equilibrium is unstable and that equilibrium must be attained?

===PREPARING YOUR MANUSCRIPT===

If you have been asked to revise the written English in your submission as a condition of publication, you must do so, and you are expected to provide evidence that you have received

language editing support. The journal would prefer that you use a professional language editing service and provide a certificate of editing, but a signed letter from a colleague who is a native speaker of English is acceptable. Note the journal has arranged a number of discounts for authors using professional language editing services (<https://royalsociety.org/journals/authors/benefits/language-editing/>).

===PREPARING YOUR REVISION IN SCHOLARONE===

Author's Response to Decision Letter for (RSOS-201808.R1)

See Appendix B.

Decision letter (RSOS-201808.R2)

Dear Mr Prieto Curiel,

I am pleased to inform you that your manuscript entitled "A paradox of traffic and extra cars in a city as a collective behaviour" is now accepted for publication in Royal Society Open Science.

on behalf of Professor Mark Chaplain (Subject Editor)
openscience@royalsociety.org

REVIEWER 1

Reviewer (R1.1): *My background is public health medicine, so I come to transport issues from a different position than the authors of this paper. In my view, the model of transport choices that is applied here leaves out many important factors that affect opportunities and decisions. But the findings are of interest, and they seem to align well with recommendations made on other grounds (eg trials of interventions, analysis of the policy landscape, case studies). The major points, as I see them, are the need for restraints on car use as well as incentives for other modes, that service quality (including comfort, security and attractiveness) are key factors in PT, and that the present transport arrangements are full of perversity, such as the bad results for everyone of increasing individual amenity (supposedly) by building more roads and producing more cars.*

Thank you very much for your comments and your suggestions. Indeed transport choices can be addressed from many angles, and we appreciate the public health perspective. We have made several amendments to the manuscript according to your insights. We have also corrected the title of our manuscript.

Reviewer (R1.2): *My suggestions for minor changes, that I think would improve the paper are: acknowledge that it is not just travel time that looms large in every commuter's mind, but the variability in travel time. And in particular how consistent and predictable travel times are. See this paper ('Why are cyclists the happiest commuters?' JTH 2019) and the finding that people who switched to e-bikes were particularly satisfied by the certainty of travel time with this mode.*

Indeed, considering only travel time leaves out many other aspects influencing travellers' choices, being the reliability one of them. Following your suggestion, we have now included in the article, Section 2(a,b), evidence on other factors influencing the mode choice of travellers.

Other more general factors are also mentioned in page 4, including the suggested reference.

Reviewer (R1.3): *I think 'selfish behaviour' is a poor choice of language. I understand I think what is meant here (choices that disregard effects on others). However, in reality, travel choices are conditioned by so many factors, outside the individual's control. Low incomes mean affordable housing for many sits on the far edge of cities, casualised work practices mean commuting is unpredictable and routes change frequently, gender roles bring, for women typically, family responsibilities that strongly influence travel decisions.*

Authors agree that there are too many factors on travel choice, many outside the individual's control. However, what we mean by 'selfish behaviour' is that given those conditions, a person tries to minimise their cost (travel time) or maximise their utility, even if that comes at the cost of the collective outcome. Similar to what is observed in Braess's paradox, what we notice is that travel choice is a setting in which the *tragedy of the commons* is observed. We have added the following text in page 13, in the discussion section, within the context of individual decision-making processes.

From economics, there is an extensive tradition in welfare theory where the tragedy of the commons and the concept of externality have a relevant role. An externality is any positive or negative effect on other individuals that are not taking into account for those who impose the effect. Therefore, it needs to be integrated into their decision process; usually, the government enforces the

inclusion of the externality into individuals' decision process through law or similar, because they would not do it by themselves [1]. Examples of negative externalities and its attempt to correct them are environmental taxes on polluting industries or cash transfers in the education or health sector. In these examples, the purely individual behaviour creates an inferior situation compared to the potential maximum social welfare [2]. Economics acknowledges congestion as a negative externality that needs to be corrected to contribute to seeking society to its maximum social welfare. Solutions in this direction have implemented pricing congestion through road toll as an effective policy [3, 4, 5, 6]. The model emphasises the need for collective behaviour using a parsimonious approach with minimal assumptions that are feasible [7, 8]. Even though individual attributes of each journey condition travel choices, we are trying to observe the emergence of a collective decision-making process to generalise from such individual aspects to an overall, city-level observation of public transport use. Similarly, from an economics point of view, lower prices of a certain good are assumed to increase demand, although many individual aspects are ignored, favouring a collective observation.

Reviewer (R1.4): *In general, I suggest the paper should acknowledge more prominently the structural and environmental factors that place severe boundaries around 'rational' personal choices. All of us are at the whim, it feels, of the industries that promote cars heavily, lobby for permissive policies and regulation, and are invested heavily in wide-spread urban form and the infrastructure that is in place. Just as an example, and I know this is not directly relevant to the subject of the paper, but it is a powerful story about whose choices matter most, take a look at 'High and Mighty. SUVs: the world's most dangerous vehicles and how they got that way' by Keith Bradsher.*

Thank you very much for the suggestion of K. Bradsher's book. We share your view that we are somehow trapped between industries that promote cars heavily and permissive policies that do not look for a common good and do not encourage social efficiency, promoting only individual choices. Also, these policies shape the cities in which we live, where car infrastructure is an intrinsic part of cities landscape around the globe. We have addressed your suggestions by adding the following paragraphs to the discussion section as well:

The rapid growth of urban population, land-use policies and car-oriented infrastructure that dominated the 20th Century derived in sprawling low-dense suburban areas, increasing the commuting distance at the expense of active modes of transportation (walk and bicycle), and making it economically costly to introduce efficient public transport with high frequency and reachable at short walking distance [9, 10]. Therefore, it is no surprise that the car became the preferred mode of transportation for many city-dwellers, thus increasing the number of motorised trips and, as a consequence, augmenting the congestion and the air pollution. Moreover, the large amount of public resources put into car infrastructure (roads, freeways, parking lots) and the bailouts paid to the car industry for the 2008 global financial crisis are subsidies that hide the real economic costs of car use, not to mention the unethical industry efforts to hide the negative environmental and health impacts of car use [11, 12, 13]. In this context, it would be difficult to claim that our choices are rational since information and environmental constraints impact our choices. Providing citizens with more travel options, local shops and services (mixed land use) can decrease car use [14, 15]. Furthermore, information about their choices' social costs can influence their travel decisions towards more socially efficient alternatives [16, 17].

Reviewer (R1.5): *Travellers are economic agents who maximise their expected utility: this is a convenient assumption for the modelling, but is such a flimsy and abbreviated picture of how the world works, I think it needs some mighty strong caveats associated with it. The mobilities literature is packed with studies (including some great papers from the TSU at Oxford!), pointing out the role of experiences such as nature contact, social interactions, physical activity, the sense of agency that accompanies personal effort, the 'nimbleness' of certain forms of travel, and as I have mentioned already, reliability of arrival time. The status associated with certain forms of travel (this is mentioned I think in the paper - and is certainly a factor in places like China where driving really is perverse in the presence of an efficient metro) and the ways in which travel asserts*

and promotes identities (eg fixie riders in big cities like London - madness, but it's a powerful statement of who you are; drivers of double cab utes likewise).

Indeed the maximising utility theory allows us to express the problem more conveniently. Yet, using expected travel time as the variable which individuals try to maximise is a strong assumption. Reliability and other aspects, such as social interactions, physical activity and many more, play a significant role in travel choice. There is a vast literature that deals with such variables in travel choice, and there are different behavioural theories proposed to explain human choice behaviour. We updated the list of factors and references (including some papers from our colleagues from TSU) to acknowledge the problem's complexity. Also, we have rewritten the paragraph about human decision theories and included a paragraph on equilibrium to better justify the use of the utility maximisation principle. It reads as follows, in page 3, section 2(a):

Travellers' choice behaviour is a process that involves psychological and cognitive mechanisms through which travellers perceive the states of the world and then make decisions accordingly [18, 19]. Many factors intervene in this process [20], including characteristics related to the trip, such as time of the day, the purpose of the trip and its length, but also attributes of each mode of transportation and the number of options available for the user, including monetary costs, how consistent and predictable travel times are, parking availability, walking distance, waiting time, frequency, number of changes, the level of satisfaction, comfort and security [21, 22, 23]. Beyond trip and travel factors, other aspects such as socio-economic and demographic characteristics of the travellers, including income, car or bike ownership, gender, age, social status and environmental and health consciousness also play a significant role [24, 25, 26, 27, 28], as well as past experiences with a mode of transport [21], transport and land-use policies, urban sprawl, subsidies, car use restrictions, public transport infrastructure, and level of service or public transport systems [27, 25, 29, 30, 31, 32]. Even weather conditions play a role [33, 34]. These factors determine the cost incurred by travellers when choosing a mode of transportation. Car or public transport ridership is driven by the users' perceived cost, including travel time [35, 36, 37, 38].

In transportation, as well as in economics, several theories have been proposed to explain how individuals make choices and to measure the weight that the different factors have on their choices. Among the many ways human decision-making has been analysed, we find the utility maximisation theory, which assumes that agents measure and compare the available alternatives through a utility function and that they will choose the option that maximises their utility [39, 40]. Therefore, providing a theoretically sound yet simple framework to model travellers' choices and understand the factors influencing their preferences by estimating random utility models [41, 42, 43]. Nonetheless, it has been found that maximum utility theory cannot reproduce human behaviour in certain circumstances, and thus alternative theories have been developed, such as bounded rationality [44, 45] and prospect theory [46, 47, 48], that try to account for cognitive biases in the decision-making process. Even though these theories provide insights on different aspects of human choice behaviour, none can be considered "true" and all have some limitations. Here, we assume that travellers are economic agents who maximise their expected utility or follow the principle of least effort [49], a common assumption among mobility models [50, 51], based on the idea that passengers try to minimise their travel cost and they choose between using a car or any other transport method.

Reviewer (R1.6): *I am intrigued by the finding that random allocation of travel mode might produce better results overall than relying on individuals to make good decisions. Does this have any practical applications? Car-free days based on the last digit on the registration plate? Free PT for those born in a particular month of the year?*

In some cities in Latin America and elsewhere, car-free days based on the registration plate has been a policy in place, without significant results. In Mexico City, for example, the policy

encouraged drivers to buy additional older and hence more polluting vehicles, increasing pollution and vehiculation in the city. Perhaps a policy based on the individual (and not on the car) could yield better results. Still, the idea of random allocation is just an extreme scenario to show that flipping a coin would sort trips more efficiently. Some cities, including Mexico City, Beijing, Athens, Quito and Bogotá, have introduced programs that ban drivers from using their vehicles [52] although the impact of such programs tends to be limited and lasting only a few months [53]. In Mexico City, for example, a policy implemented in 1989 banned vehicles based on their license plate, but the policy led to an increase in the total number of cars [52]. In the long run, the policy encouraged drivers to buy additional older and hence more polluting vehicles to circumvent the restrictions [54]. Convincing drivers of avoiding single-occupancy vehicle commutes, using public transport, cycling, and walking cannot rely on a single policy but on increasing journey options, accessibility, and time efficiency.

We have added a comment in page 13, in the discussion of the manuscript.

Thank you for your valuable inputs on our manuscript.

 REVIEWER 2

Reviewer (R2.1): *The paper deals with a problem of city planning regarding mode of transportation and what modes can lead to better commuting times. The paper defines an equilibrium point but I'm not certain the assumptions are realistic enough for the paper to be very useful in practice.*

Thank you very much for your comments and your suggestions. Our results are based on strong assumptions, some of which have been addressed based on your comments and the suggestions of Reviewer 1. We think our results point to some relevant aspects of public transport that could be tackled as a mobility policy in the city, mainly when users are observed at a collective level and not individually. Our model cannot reproduce individual aspects but gives an insight into how even a paradox might be observed, where all individuals try to minimise travel time but collectively, they reach the worst-case scenario with the highest travel time. This result is similar to other economic models where it is considered that lower prices of a certain good are assumed to increase its demand. However, many individual aspects are being ignored in favour of a model which explains a collective observed result of higher demand.

Reviewer (R2.2): *The issue with the use of motorised means of transportation is not a uniform issue. Clearly the USA has such issues but the same is not true in many European cities. In fact there is evidence of a huge increase in the use of no-motorised modes of transportation (bike shares) in New York, Beijing, Philadelphia (with numbers ranging from 70% to 150% increases). It would be nice if the authors could contrast their results with what is happening in real cities. The analytical analysis could be contextualised with real data. Do cities reach this equilibrium cost?*

As you mention, motorisation is not uniform across cities. The built environment and other social aspects have a strong influence on commuters' choices and, as a consequence, the modal share will be different for different cities. Furthermore, these factors are not static, explaining the changing conditions of travel in the same urban areas. To address your comment concerning equilibrium, we include the following paragraphs in the article in page 3, section 2(a), as a follow-up to the answer to comment R1.5:

Equilibrium depends on which behavioural theory and which factors are considered. We can theoretically assume a network to be in equilibrium if every individual is trying to maximise their satisfaction (unknown for us) by means of their (imperfect) perception of the world states and the limitations of their environment. In practice, establishing if a city is in equilibrium is more complex, as our observations about the decision-makers and the choice situations are limited. In idealised scenarios, such as laboratory decision experiments, it has been found that equilibrium is reached after repeated choices of participants [55, 56]. However, establishing if an equilibrium is reached at a city scale requires knowing the weight that the different factors have in the preference of the decision-makers and the states of these factors when people made their choices. Since travel time is a variable that can be measured in a real-world setting (through GPS traces, for example), in practice, it is often assumed that it is the only variable that intervenes in the decisions of travellers. In the context of route choice, for example, it has been found that at an individual level, car drivers do not follow the fastest path, but that the traffic states of the network are close to the theoretical user equilibrium, which considers that travellers are travel time minimisers [57]. Furthermore, even though travellers do not follow the fastest path, there is a

threshold for the extra time they are willing to incur by choosing sub-optimal routes [58, 59, 60, 61, 62].

If we assume that a transport system reaches equilibrium (in the morning peak hour, for example), then the trips' modal share is in equilibrium. Yet, a change in the factors that influence travellers' decisions, or a change in their preferences, will move the stability. This has been observed, for example, in mobility reductions [63], fewer users of public transport [64] and people who now find walking and cycling more attractive as a result of the COVID-19 pandemic [65].

Reviewer (R2.3): *I found Figure 1 quite hard to understand. For instance, why does the dashed line for public transportation cost with no car change as a function of car users (if there are no cars)?*

In the model, we consider that the travel time of cars and public transport is a function of the number of users in both systems. The costs are assumed to be linear so that the travel time of car users increases with more car users (dashed line), but also, there are indirect costs from public transport users (such as lanes devoted to public transport, queues, space occupancy and others). Since we assume that all people choose between the two modes of transport (car and others), then more car users imply extra time due to the direct costs of the extra drivers, but also, a small reduction in the indirect costs due to fewer public transport users. The combined total cost (solid line) increases with the number of car users. With more car users, there are less indirect costs from public transport (so fewer lanes devoted to public transport, queues, space occupancy and others). That is why, with sufficient cars, the dashed and the solid line intersect. Mathematically speaking, under the linear cost assumptions, we get that the cost time (in minutes) for car users is $C_0 + \alpha C + \beta T$, where C is car users, T is transport users, and the total cost is the dashed line in Figure 1. Notice that there are two components, $C_0 + \alpha C$, which is the dashed line, and represents the costs of using cars, and a second component βT , based on the number of transport users. Since all people use a system, then $T = N - C$ from which we can rewrite the total cost for car users in minutes as $C_0 + \beta N + (\alpha - \beta)C$, the solid line.

We have changed the figure and the paragraph, introducing the equations to make it clearer.

Reviewer (R2.4): *The paper is an interesting exercise but without evidence of such modelling in real cities, the paper becomes less relevant. Sure it is an interesting modelling but how can we know this model fits or can explain what we see in real cities? At least a simple indication would be needed. I suggest a case study using the data for a couple of cities with different scenarios of public transportation and number of cars (e.g. Miami vs. London)*

Recently it was estimated, using Twitter data, that using public transport in four cities (São Paulo, Sydney, Stockholm, and Amsterdam) takes, on average, between 1.4 and 2.6 times longer than driving a car ([38]), although the ratio is usually smaller for short distances. Their results suggest that users are willing to have an extra commuting time for long distances when using public transport of up to twice the driving time. In turn, the fact that there is a roughly similar travel time ratio suggests two things. Firstly, as with your previous comment, cities reach a transport equilibrium, where some aspects of public transport and active mobility compensate for the fact that they take twice as long for a regular commute. But also, the stable ratio indicates that with extra travel time on a car (due to the excess number of vehicles, the rush hour or some policies, like reducing the space in cities devoted to cars and reducing parking), people will likely choose to use the public transport instead.

We have added a paragraph in section 2(c), page 5, to highlight the equilibrium according to your comments.

Reviewer (R2.5): *The literature review is short but perhaps adequate but it misses important works such as the recent works on mobility models.*

Based on your suggestions and the suggestions by Reviewer 1, we have added some points to our literature review section, particularly concerning the different reasons why a person picks a

transport mode and the possible implications of various programs. We have also added some recent manuscripts, including some which mention the impact of COVID on transport use. We hope that this covers the many aspects of transport mode and highlights to the readers the vast and complex research area that transportation is.

Reviewer (R2.6): *There are many grammatical mistakes and typos in the document. It should be carefully proof-read before submitting it again.*

Thank you for your valuable suggestions. We have carefully proofread the manuscript and modify many aspects accordingly.

REFERENCES

- [1] Baumol, W.J.: *Welfare Economics and the Theory of the State*. Springer (2004)
- [2] Marciano, A., Frischmann, B.M., Ramello, G.B.: *Tragedy of the commons after 50 years*. Available at SSRN 3451688 (2019)
- [3] Brinkman, J.C.: *Congestion, agglomeration, and the structure of cities*. *Journal of Urban Economics* **94**, 13–31 (2016)
- [4] Lindsney, R., Verhoef, E.: *Traffic Congestion and Congestion Pricing*. Emerald Group Publishing Limited, Amsterdam, Netherlands (2001)
- [5] Parry, I.W., Walls, M., Harrington, W.: *Automobile externalities and policies*. *Journal of Economic Literature* **45**(2), 373–399 (2007)
- [6] Russo, A., Adler, M., Liberini, F., van Ommeren, J.N.: *Welfare losses of road congestion* (2019)
- [7] Couclelis, H.: *From sustainable transportation to sustainable accessibility: Can we avoid a new tragedy of the commons?* In: *Information, Place, and Cyberspace*, pp. 341–356. Springer, Berlin, Heidelberg (2000)
- [8] López, L., del Rey Almansa, G., Paquelet, S., Fernández, A.: *A mathematical model for the TCP tragedy of the commons*. *Theoretical Computer Science* **343**(1-2), 4–26 (2005)
- [9] Marchetti, C.: *Anthropological invariants in travel behavior*. *Technological Forecasting and Social Change* **47**(1), 75–88 (1994). doi:10.1016/0040-1625(94)90041-8
- [10] Mattioli, G., Roberts, C., Steinberger, J.K., Brown, A.: *The political economy of car dependence: A systems of provision approach*. *Energy Research & Social Science* **66**, 101486 (2020). doi:10.1016/j.erss.2020.101486
- [11] Bradsher, K.: *High and Mighty: The Dangerous Rise of the SUV*. PublicAffairs, New York (2004)
- [12] Chossière, G.P., Malina, R., Ashok, A., Dedoussi, I.C., Eastham, S.D., Speth, R.L., Barrett, S.R.H.: *Public health impacts of excess NOx emissions from Volkswagen diesel passenger vehicles in Germany*. *Environmental Research Letters* **12**(3), 034014 (2017). doi:10.1088/1748-9326/aa5987
- [13] Oldenkamp, R., van Zelm, R., Huijbregts, M.A.J.: *Valuing the human health damage caused by the fraud of Volkswagen*. *Environmental Pollution* **212**, 121–127 (2016). doi:10.1016/j.envpol.2016.01.053
- [14] Giuliano, G., Dargay, J.: *Car ownership, travel and land use: a comparison of the US and Great Britain*. *Transportation Research Part A: Policy and Practice* **40**(2), 106–124 (2006). doi:10.1016/j.tra.2005.03.002
- [15] Zhao, P.: *Car use, commuting and urban form in a rapidly growing city: evidence from Beijing*. *Transportation Planning and Technology* **34**(6), 509–527 (2011). doi:10.1080/03081060.2011.600049. <https://doi.org/10.1080/03081060.2011.600049>
- [16] van Essen, M., Thomas, T., van Berkum, E., Chorus, C.: *From user equilibrium to system optimum: a literature review on the role of travel information, bounded rationality and non-selfish behaviour at the network and individual levels*. *Transport Reviews* **36**(4), 527–548 (2016)
- [17] Mariotte, G., Leclercq, L., Gonzalez Ramirez, H., Krug, J., Bécarie, C.: *Assessing traveler compliance with the social optimum: A stated preference study*. *Travel Behaviour and Society* **23**, 177–191 (2021). doi:10.1016/j.tbs.2020.12.005
- [18] Bovy, P.H.L., Stern, E.: *Route Choice: Wayfinding in Transport Networks*. Kluwer Academic Publishers, Netherlands (1990)
- [19] Ben-Akiva, M., McFadden, D., Gärling, T., Gopinath, D., Walker, J., Bolduc, D., Börsch-Supan, A., Delquié, P., Larichev, O., Morikawa, T., Polydoropoulou, A., Rao, V.: *Extended Framework for Modeling Choice Behavior*. *Marketing Letters* **10**(3), 187–203 (1999)
- [20] Buehler, R.: *Determinants of transport mode choice : a comparison of Germany and the USA*. *Journal of Transport Geography* **19**(4), 644–657 (2011)
- [21] De Vos, J., Schwanen, T., Van Acker, V., Witlox, F.: *Do satisfying walking and cycling trips result in more future trips with active travel modes? an exploratory study*. *International Journal of Sustainable Transportation* **13**(3), 180–196 (2019)
- [22] Wild, K., Woodward, A.: *Why are cyclists the happiest commuters? health, pleasure and the e-bike*. *Journal of Transport & Health* **14**, 100569 (2019)

- [23] Kuhnimhof, T., Buehler, R., Wirtz, M., Kalinowska, D.: Travel trends among young adults in Germany : increasing multimodality and declining car use for men. *Journal of Transport Geography* **24**, 443–450 (2012)
- [24] Hopkins, D., García Bengoechea, E., Mandic, S.: Adolescents and their aspirations for private car-based transport. *Transportation*, 1–27 (2019)
- [25] Bautista-Hernández, D.A.: The urban form and the social dimension of commuting in Mexico City. an individual trip-level analysis. *Transportation Research Interdisciplinary Perspectives* **10**, 100346 (2021). doi:10.1016/j.trip.2021.100346
- [26] Wu, T., Zhao, H., Ou, X.: Vehicle ownership analysis based on GDP per capita in China: 1963–2050. *Sustainability* **6**(8), 4877–4899 (2014)
- [27] Gascon, M., Marquet, O., Gràcia-Lavedan, E., Ambròs, A., Götschi, T., de Nazelle, A., Panis, L.I., Gerike, R., Brand, C., Dons, E., Eriksson, U., Iacorossi, F., Àvila-Palència, I., Cole-Hunter, T., Nieuwenhuisjen, M.J.: What explains public transport use? evidence from seven European cities. *Transport Policy* **99**, 362–374 (2020). doi:10.1016/j.tranpol.2020.08.009
- [28] Mandic, S., Hopkins, D., García Bengoechea, E., Flaherty, C., Coppel, K., Moore, A., Williams, J., Spence, J.C.: Differences in parental perceptions of walking and cycling to high school according to distance. *Transportation Research Part F: Traffic Psychology and Behaviour* **71**, 238–249 (2020). doi:10.1016/j.trf.2020.04.013
- [29] Cervero, R.: Built environments and mode choice: toward a normative framework. *Transportation Research Part D: Transport and Environment* **7**(4), 265–284 (2002)
- [30] Ewing, R., Cervero, R.: Travel and the built environment: A meta-analysis. *Journal of the American Planning Association* **76**(3), 265–294 (2010)
- [31] Li, P., Zhao, P., Schwanen, T.: Effect of land use on shopping trips in station areas: Examining sensitivity to scale. *Transportation Research Part A: Policy and Practice* **132**, 969–985 (2020). doi:10.1016/j.tra.2019.12.029
- [32] Chan, E.T.H., Li, T.E., Schwanen, T., Banister, D.: People and their walking environments: An exploratory study of meanings, place and times. *International Journal of Sustainable Transportation* **0**(0), 1–12 (2020). doi:10.1080/15568318.2020.1793437. <https://doi.org/10.1080/15568318.2020.1793437>
- [33] Abad, R.P.B., Schwanen, T., Fillone, A.M.: Commuting behavior adaptation to flooding: An analysis of transit users’ choices in Metro Manila. *Travel Behaviour and Society* **18**, 46–57 (2020). doi:10.1016/j.tbs.2019.10.001
- [34] Budnitz, H., Tranos, E., Chapman, L.: Responding to stormy weather: Choosing which journeys to make. *Travel Behaviour and Society* **18**, 94–105 (2020). doi:10.1016/j.tbs.2019.10.008
- [35] De Vos, J., Mokhtarian, P.L., Schwanen, T., Van Acker, V., Witlox, F.: Travel mode choice and travel satisfaction: bridging the gap between decision utility and experienced utility. *Transportation* **43**(5), 771–796 (2016)
- [36] Beirão, G., Cabral, J.S.: Understanding attitudes towards public transport and private car: A qualitative study. *Transport Policy* **14**(6), 478–489 (2007)
- [37] Eriksson, L., Friman, M., Gärling, T.: Stated reasons for reducing work-commute by car. *Transportation Research Part F: Traffic Psychology and Behaviour* **11**(6), 427–433 (2008)
- [38] Liao, Y., Gil, J., Pereira, R.H., Yeh, S., Verendel, V.: Disparities in travel times between car and transit: Spatiotemporal patterns in cities. *Scientific Reports* **10**(1), 1–12 (2020)
- [39] Wardrop, J.G.: Road paper. some theoretical aspects of road traffic research. *Proceedings of the Institution of Civil Engineers* **1**(3), 325–362 (1952). doi:10.1680/ipeds.1952.11259
- [40] Daganzo, C.F., Sheffi, Y.: On stochastic models of traffic assignment. *Transportation Science* **11**(3), 253–274 (1977). doi:10.1287/trsc.11.3.253
- [41] Marschak, J.: Binary choice constraints on random utility indicators. Cowles Foundation Discussion Papers 74, Cowles Foundation for Research in Economics, Yale University (1959)
- [42] Mcfadden, D.: Conditional logit analysis of qualitative choice behavior. In: Zarembka, P. (Ed.), *Frontiers of Econometrics* (1972)
- [43] McFadden, D., Train, K.: Mixed MNL models for discrete response. *Journal of Applied Econometrics* **15**(5), 447–470 (2000)
- [44] Simon, H.A.: *Models of Man Social and Rational : Mathematical Essays on Rational Human Behavior in a Social Setting*, p. 287. Wiley New York, USA (1957)
- [45] Mahmassani, H.S., Chang, G.-L.: On boundedly rational user equilibrium in transportation systems. *Transportation Science* **21**(2), 89–99 (1987). doi:10.1287/trsc.21.2.89
- [46] Kahneman, D., Tversky, A.: Prospect Theory: An Analysis of Decision under Risk. *Econometrica: Journal of the Econometric Society* **47**(3), 263–291 (1979). 9809069v1
- [47] Xu, H., Zhou, J., Xu, W.: A decision-making rule for modeling travelers’ route choice behavior based on cumulative prospect theory. *Transportation Research Part C: Emerging Technologies* **19**(2), 218–228 (2011). doi:10.1016/j.trc.2010.05.009. Emerging theories in traffic and transportation and methods for transportation planning and operations
- [48] De Moraes Ramos, G., Daamen, W., Hoogendoorn, S.: Modelling travellers’ heterogeneous route choice behaviour as prospect maximizers. *Journal of Choice Modelling* **6**, 17–33 (2013)

- [49] Zipf, G.K.: Human behavior and the principle of least effort (1949)
- [50] Guy, S.J., Chhugani, J., Curtis, S., Dubey, P., Lin, M.C., Manocha, D.: PLEdestrians: A least-effort approach to crowd simulation. In: Symposium on Computer Animation, pp. 119–128 (2010)
- [51] Al-Widyan, F., Kirchner, N., Zeibots, M.: An empirically verified passenger route selection model based on the principle of least effort for monitoring and predicting passenger walking paths through congested rail station environments. In: Australasian Transport Research Forum 2015 Proceedings, pp. 1–12 (2015)
- [52] Davis, L.W.: The effect of driving restrictions on air quality in Mexico City. *Journal of Political Economy* **116**(1), 38–81 (2008)
- [53] Gallego, F., Montero, J.-P., Salas, C.: The effect of transport policies on car use: Evidence from Latin American cities. *Journal of Public Economics* **107**, 47–62 (2013)
- [54] Carrillo, P.E., Malik, A.S., Yoo, Y.: Driving restrictions that work? Quito’s pico y placa program. *Canadian Journal of Economics/Revue Canadienne d’Économie* **49**(4), 1536–1568 (2016)
- [55] Selten, R., Chmura, T., Pitz, T., Kube, S., Schreckenberg, M.: Commuters route choice behaviour. *Games and Economic Behavior* **58**(2), 394–406 (2007). doi:10.1016/j.geb.2006.03.012
- [56] Rapoport, A., Kugler, T., Dugar, S., Gisches, E.J.: In: Kugler, T., Smith, J.C., Connolly, T., Son, Y.-J. (eds.) *Braess Paradox in the Laboratory: Experimental Study of Route Choice in Traffic Networks with Asymmetric Costs*, pp. 309–337. Springer, New York, NY (2008)
- [57] Yildirimoglu, M., Kahraman, O.: Searching for empirical evidence on traffic equilibrium. *PLOS ONE* **13**(5), 1–16 (2018). doi:10.1371/journal.pone.0196997
- [58] Bekhor, S., Ben-Akiva, M.E., Ramming, M.S.: Evaluation of choice set generation algorithms for route choice models. *Annals of Operations Research* **144**(1), 235–247 (2006)
- [59] Papinski, D., Scott, D.M., Doherty, S.T.: Exploring the route choice decision-making process: A comparison of planned and observed routes obtained using person-based gps. *Transportation Research Part F: Traffic Psychology and Behaviour* **12**(4), 347–358 (2009)
- [60] Thomas, T., Tutert, S.: Route choice behavior based on license plate observations in the Dutch city of Enschede. In: Seventh Triennial Symposium on Transportation Analysis: Tristan VII, Tristan (2010)
- [61] Zhu, S., Levinson, D.: Do people use the shortest path? An empirical test of Wardrop’s first principle. *PLoS ONE* **10**(8), 1–18 (2015)
- [62] Hadjidimitriou, S.N., Dell’Amico, M., Cantelmo, G., Viti, F.: Assessing the consistency between observed and modelled route choices through gps data. In: 2015 International Conference on Models and Technologies for Intelligent Transportation Systems (MT-ITS), pp. 216–222 (2015). doi:10.1109/MTITS.2015.7223259
- [63] Lee, W., Qian, M., Schwanen, T.: The association between socioeconomic status and mobility reductions in the early stage of England’s COVID-19 epidemic. *Health and Place* (2021)
- [64] Teixeira, J.F., Lopes, M.: The link between bike sharing and subway use during the COVID-19 pandemic: the case-study of New York’s citi bike. *Transportation Research Interdisciplinary Perspectives* **6**, 100166 (2020)
- [65] Nikiforiadis, A., Ayfantopoulou, G., Stamelou, A.: Assessing the impact of COVID-19 on bike-sharing usage: The case of Thessaloniki, Greece. *Sustainability* **12**(19), 8215 (2020)

Appendix B

Point by point answer

REVIEWER 1

Reviewer (R1.1): Thank you for the chance to read this paper again. My questions about the first version, and suggestions of possible improvements, have been treated thoughtfully, and the paper is stronger in its present form. I think it warrants publication as is.

Thank you very much for your valuable inputs on our manuscript.

REVIEWER 2

Reviewer (R2.1): *The paper is at a much better state than before and I have just a couple of comments still. Figure 1 continues to be confusing IMO. I understand it may not be possible to change it but could it be broken down into two parts? I think the authors are trying to convey too much in one figure.*

Thank you very much for your comments and your suggestions. We also feel that this new version has a clearer message for the readers. We modified figure 1 and figure 3, trying to obtain a cleaner version, with less equations and symbols in it.

Reviewer (R2.2): *I'm still slightly disappointed about the applicability of the results in city planning, city modelling, etc. A stronger discussion session where the authors address the "so what" would be useful. The authors mention in the response that cities appear to reach equilibrium, so can I reverse the question? Are there cities that are not in equilibrium? Can you give an intuition as whether this non-equilibrium is unstable and that equilibrium must be attained?*

Whilst a short-term solution to reduce the price of anarchy would be to only allow some cars on the street based on their license (or, as it was enforced in some parts, due to the COVID pandemic and not to reduce congestion or pollution) or based on some individual aspects, such as gender (as it happened in Bogotá) or the national ID (as it happened in Medellín) the system eventually became unsustainable.

Cities have severe shocks, for example, recently, a Metro line in Mexico City collapsed, with 26 casualties ¹ leaving thousands of people disconnected from the main metro system. Unfortunately, the disaster occurred in some impoverished parts of the city with very long commutes to the working and commercial areas. The shock will be suffered by citizens for months whilst they adjust to reduced mobility options in those neighbourhoods. Still, people will find a way to go back home. The shock brings a new equilibrium to the city, with longer commutes and fewer options in the area. But one of the biggest disasters is also that public transport users have a new reason why they should avoid public transport in Mexico City. Not only is insecure for women, takes longer to get there and is crowded and less comfortable, but also, public transport users now have to fear a new disaster happening. Thus, we argue that instead of fighting to reduce the price of anarchy by a collective way of decision-making (such as flipping a coin), we need to plan cities so that the price of anarchy is lower. We need fewer journeys, shorter distances, accessibility so that people can walk, cycling lanes and more public transport, so that the best transport mode for everyone is not to use a car.

¹<https://www.bbc.com/news/world-latin-america-56977129>